# Transposon-Directed Insertion-Site Sequencing Reveals Glycolysis Gene *gpmA* as Part of the H_2_O_2_ Defense Mechanisms in *Escherichia coli*

**DOI:** 10.3390/antiox11102053

**Published:** 2022-10-18

**Authors:** Myriam Roth, Emily C. A. Goodall, Karthik Pullela, Vincent Jaquet, Patrice François, Ian R. Henderson, Karl-Heinz Krause

**Affiliations:** 1Department of Pathology and Immunology, Faculty of Medicine, University of Geneva, 1211 Geneva, Switzerland; 2Institute for Molecular Bioscience, University of Queensland, St. Lucia, Brisbane, QLD 4072, Australia; 3READS Unit, Faculty of Medicine, University of Geneva, 1211 Geneva, Switzerland; 4Genomic Research Laboratory, Infectious Diseases Service, University Hospitals of Geneva, University Medical Center, Michel-Servet 1, 1211 Geneva, Switzerland

**Keywords:** *E. coli*, H_2_O_2_, TraDIS, Tn-seq, phophoglycerate mutase, *gpmA*

## Abstract

Hydrogen peroxide (H_2_O_2_) is a common effector of defense mechanisms against pathogenic infections. However, bacterial factors involved in H_2_O_2_ tolerance remain unclear. Here we used transposon-directed insertion-site sequencing (TraDIS), a technique allowing the screening of the whole genome, to identify genes implicated in H_2_O_2_ tolerance in *Escherichia coli*. Our TraDIS analysis identified 10 mutants with fitness defect upon H_2_O_2_ exposure, among which previously H_2_O_2_-associated genes (*oxyR*, *dps*, *dksA*, *rpoS*, *hfq* and *polA*) and other genes with no known association with H_2_O_2_ tolerance in *E. coli* (*corA*, *rbsR*, *nhaA* and *gpmA*). This is the first description of the impact of *gpmA*, a gene involved in glycolysis, on the susceptibility of *E. coli* to H_2_O_2_. Indeed, confirmatory experiments showed that the deletion of *gpmA* led to a specific hypersensitivity to H_2_O_2_ comparable to the deletion of the major H_2_O_2_ scavenger gene *katG*. This hypersensitivity was not due to an alteration of catalase function and was independent of the carbon source or the presence of oxygen. Transcription of *gpmA* was upregulated under H_2_O_2_ exposure, highlighting its role under oxidative stress. In summary, our TraDIS approach identified *gpmA* as a member of the oxidative stress defense mechanism in *E. coli*.

## 1. Introduction

*Escherichia coli* is a Gram-negative facultative anaerobic bacterium. It is a frequent member of the normal human microbiota but can also be a pathogen causing food poisoning, urinary tract infection and even septic shock [1]. The burden of diarrheal infections by pathogenic strains of *E. coli* is immense; in 79 low-income countries alone, more than 200 million episodes of childhood diarrhea due to *E. coli* and *Shigella* occur each year [2]. In high-income countries, *E. coli* is the primary cause of blood stream infections, accounting for 27% of the documented bacteremia episodes [3]. The emergence of antibiotic resistance in Gram-negative bacteria is also concerning and a recent study of 203 countries identified *E. coli* as the leading pathogen for deaths associated with antimicrobial resistance in 2019 [4].

Reactive oxygen species (ROS), and more specifically hydrogen peroxide (H_2_O_2_), have a strong impact on bacterial pathogenesis. Millimolar H_2_O_2_ can be produced by certain strains of *Lactobacilli* of the human normal microbiota [5]. H_2_O_2_ production prevents the colonization by pathogens of the urinary tract [6]. Similarly, H_2_O_2_ is produced by phagocytes during the oxidative burst, a necessary step for the killing of pathogens [7]. The effect of H_2_O_2_ on bacteria has been partially studied, but a complete picture of how H_2_O_2_ affects bacteria and the bacterial response has not been elucidated for any bacterial species. Previous studies on H_2_O_2_ tolerance, using DNA microarrays and RNA-seq, identified genes regulated under H_2_O_2_ exposure [8,9,10] in *E. coli*. These studies permitted a better understanding of the regulation of numerous genes and pathways affected by H_2_O_2_ exposure. In particular, OxyR, a specific H_2_O_2_-responsive transcription factor, and SoxR which senses oxidative stress and nitric oxide, have been identified as playing an important role in resistance to H_2_O_2_ [9,11]. OxyR senses hydrogen peroxide through the oxidation of its cysteine residues, which orchestrate a conformational change allowing it to regulate the expression of 38 genes [9,12]. The iron–sulfur cluster of SoxR is oxidized by redox cycling compounds or superoxide, leading to the activation of the transcription factor which regulates the expression of 11 genes, which includes SoxS, another transcription factor that further regulates 34 genes [9,13]. However, transcriptional analyses do not identify genes required for survival in oxidative conditions. Diverse mutagenesis techniques were used to identify the genes involved in *E. coli* survival under H_2_O_2_ exposure, but only a limited number of genes were identified each time [14,15,16].

The combination of transposon mutagenesis and high-throughput sequencing is a powerful technique that allows interrogation of the whole genome and represents a new standard for global functional genomic studies [17]. Here, we performed transposon-directed insertion-site sequencing (TraDIS) [18] to identify genes implicated in tolerance to exogenous H_2_O_2_ exposure. A similar approach was used on *Salmonella* Typhimurium to identify genes implicated in H_2_O_2_ tolerance, deepening the understanding of how the bacteria survive oxidative burst [19,20]. The results of our study highlighted the role of *gpmA,* which encodes a phosphoglycerate mutase, an enzyme of the glycolysis, under H_2_O_2_ exposure. This is the first study identifying *gpmA* as a factor of H_2_O_2_ tolerance in *E. coli*.

## 2. Materials and Methods

### 2.1. Bacterial Strains, Media and Growth Conditions

All bacterial strains and plasmid used in this study are documented in Table 1. *E. coli* strains were cultured at 37 °C in Luria–Bertani (LB) (Becton & Dickinson, Basel, Switzerland) broth or on Luria–Bertani Agar (Becton & Dickinson). H_2_O_2_ 35% *w/w* (Acros Organics, VWR Life Science, Nyon, Switzerland) was added at the indicated final concentration. LB was supplemented with 0.4% glucose, 0.4% glycerol, 0.4% sodium acetate, 0.4% sodium citrate, 50 mM sodium nitrate (Sigma-Aldrich, St. Louis, MI, USA, Merck and Cie, Schaffhausen, Switzerland) where indicated. Minimal medium M9 plates, constituted by M9 salts (VWR Life Science), 0.1 mM CaCl_2_ (Sigma-Aldrich), 0.2 mM MgSO_4_ (Sigma-Aldrich), 1.5% (*w/v*) agar (Carl Roth, Arlesheim, Switzerland), were used when indicated. Antibiotics were used when indicated at the following concentrations: ampicillin 100 μg/mL (10044, Sigma-Aldrich), kanamycin 50 μg/mL (PanReac AppliChem, VWR, Switzerland). For anaerobic assays, bacteria were grown in deoxygenated LB with the corresponding antibiotic and every step was performed under anaerobic condition (Coy Laboratory Products, Labgene scientific SA, Châtel-Saint-Denis, Switzerland). Overnight cultures and agar plates were grown overnight at 40 °C in anaerobic chamber.

The TraDIS screen was performed using a library of transposon mutants previously generated in *E. coli* strain BW25113 [18]. The *E. coli* strain MG1655 is referred in this paper as the wild-type (WT). The strain BEFB02 with *oxyR* deletion was a kind gift from B. Ezraty. Other gene deletions were obtained from the Keio collection [21] and transduced in a MG1655 background by P1 transduction.

### 2.2. TraDIS

The TraDIS library was thawed and diluted in 50 mL of LB broth to reach an OD_595_ of 0.02 (approximatively 8 × 10^8^ CFU). H_2_O_2_ was added to H_2_O_2_-treated samples to reach a concentration of 2.5 mM whereas pure medium was added in untreated controls. The experiment was performed in duplicates. Bacteria were grown at 37 °C in aerating conditions (250 mL flask, shaking 250 rpm) until an OD_595_ = 1.

Bacteria were collected and the DNA was extracted using a DNeasy Blood & Tissue Kit (Qiagen) according to the manufacturer’s instructions. Samples were prepared for sequencing as described previously [18]. Briefly, genomic DNA was fragmented by ultrasonication, fragments were end-repaired using the NEBNext Ultra I kit (New England Biolabs, Notting Hill, Australia) and transposon fragments enriched by PCR using primers specific for the transposon and adapter. Samples were quantified by qPCR using the NEBNext Library Quant Kit for Illumina kit (New England Biolabs) according to the manufacturer’s instructions and sequenced using an Illumina MiSeq with 150-cycle v3 cartridges.

The TraDIS data were analyzed using Bio::TraDIS pipeline [24] with the following parameters: 50 reads per gene as minimal threshold and 5% trim at each side of gene to avoid the consideration of meaningless transposon insertions that can occur within gene extremities. Sequencing reads were mapped to the *E. coli* BW25113 reference genome (accession: CP009273.1) downloaded from NCBI. Sequencing reads are available at the European Nucleotide Archive (ENA) at EMBL-EBI under accession number PRJEB56340 (https://www.ebi.ac.uk/ena/browser/view/PRJEB56340, accessed on 13 October 2022). Processed data are available for viewing at our online browser: https://tradis-vault.qfab.org/.

### 2.3. P1 Transduction

Strains from the Keio library were grown with 50 mg/mL kanamycin. The deletions of genes of interest from the corresponding Keio library mutant were transduced in *E. coli* MG1655 using phage P1 as previously described [25]. P1 phage was a kind donation from G. Panis (University of Geneva). The deleted mutants were verified using PCR with appropriate gene-specific primers (Appendix A).

### 2.4. H_2_O_2_ Susceptibiliy Assessed by Disk Diffusion Assay

To assess the susceptibility to H_2_O_2_ and other oxidants, we used disk diffusion assay as previously described [10]. Briefly, an overnight culture of bacteria was diluted in LB to McFarland 0.5 using a Densimat (bioMérieux, Marcy-l’Étoile, France) and LB agar plates were inoculated using a sterile cotton swab. Sterile cellulose disks (5 mm diameter) were placed on the plate and 10 μL of 1 M H_2_O_2_ diluted in sterile water was added to the center of the disk. Other oxidants were used at the following concentrations: methylhydroquinone (Sigma-Aldrich) MHQ 0.5 M in water; methyl viologen dichloride hydrate, also called paraquat, (Sigma-Aldrich) PQ 1 M in water; diamide (Sigma-Aldrich) DI 0.2 M in water; menadione (Sigma-Aldrich) K3 360 mM in DMSO; cumene hydroperoxide (Sigma-Aldrich) CHP 0.25 M in DMSO; sodium hypochlorite (Sigma-Aldrich) NaOCl 5%; ciprofloxacin (Sigma-Aldrich) CIP 0.5 µg/µL in water; ampicillin AMP 1µg/µL in water.

Plates were incubated at 37 °C for 18 h and the diameter of inhibition was measured in mm. The area of inhibition was calculated as: [diameter of inhibition/2]^2^ × 3.14. To compare the effect of different oxidants, data were normalized as following: [area of inhibition of the interested mutant] × 100/[area of inhibition of the WT].

### 2.5. H_2_O_2_ Survival Assay

For survival assay, the susceptibility of *E. coli* to H_2_O_2_ was tested in liquid medium. Briefly, overnight cultures were diluted to 2 × 10^7^ CFU/mL in 10 mL LB. 1 mL of H_2_O_2_ diluted in LB was added to the bacterial suspension to reach a final concentration of 2.5 mM. The corresponding control received 1 mL LB without H_2_O_2_. Bacteria were grown at 37 °C, 180 rpm. At indicated time points, 20 μL of each sample were serially diluted in LB by 10-fold dilutions. 10 μL of each dilution were spotted on LB agar supplemented with 100 U/mL of bovine liver catalase (Sigma-Aldrich). Plates were incubated overnight at 37 °C. Percent survival was calculated as [CFU from H_2_O_2_-treated sample/CFU from untreated sample] × 100.

### 2.6. Expression Levels Assessed by qRT-PCR

Overnight cultures were diluted in 10 mL of LB to OD_595_ 0.02. These fresh cultures were grown at 37 °C, 180 rpm for 2 h to reach exponential phase. Bacterial suspension was divided in 2 mL samples, and 200 μL of H_2_O_2_ diluted in LB was added to reach the final concentrations indicated in the figures. The same volume of LB was added in the corresponding control conditions. Samples were incubated at 37 °C for 10 min. Subsequently, 1 mL was stabilized with 2 mL RNAprotect Bacteria Reagent (Qiagen, Hombrechtikon, Switzerland). RNA was purified using RNeasy Plus Mini Kit (Qiagen) according to the manufacturer instructions with on-column DNA digestion by RNase-Free DNase Set (Qiagen).

Quantitative PCR (qRT-PCR) was performed on RNA samples as previously described [10]. Briefly, the cDNA was produced by reverse-transcribing 500 ng of total RNA using a mix of random hexamers and oligo d(T) primers and Primescript reverse transcriptase enzyme (Takara Bio, Saint Germain-en-Laye, France). The efficiency of each pair of primers was tested with four serial dilutions of cDNA. Oligonucleotides are indicated in Table 2. PCR reactions (10 µL volume) contained 1:20 diluted cDNA, 2 × Power SYBR Green Master Mix (Thermo Fisher, Fisher Scientific AG, Reinach. Switzerland), and 300 nM of forward and reverse primers. PCRs were performed on a SDS 7900 HT instrument (Thermo Fisher) with the following parameters: 50 °C for 2 min, 95 °C for 10 min, and 45 cycles of 95 °C for 15 s, 60 °C for 1 min. Each reaction was performed in three replicates on 384-well plate. Raw Ct values obtained with SDS 2.2 (Thermo Fisher) were imported into Excel and normalization factors were calculated using the GeNorm method as described by Vandesompele et al. [26]. The absence of residual genomic DNA in RNA samples was verified by performing PCR reactions without RTase with the primer pair *gyrB*_N. Significance was assessed by one-way ANOVA with ad hoc Tukey’s multiple comparisons test.

### 2.7. H_2_O_2_ Degradation Mesurements by Amplex Red

Overnight cultures were diluted in LB to McFarland 1.0 using a Densimat (bioMérieux) and further diluted 10 fold in fresh LB. 10 mL were grown in a Falcon 50 at 37 °C for 2 h to reach exponential phase of growth. Pellets were washed with DPBS (Gibco Thermo Fisher) and resuspended to reach OD_595_ = 0.1 in DPBS. 1 mL of H_2_O_2_ diluted in sterile water was added to 10 mL of bacterial suspension for a final concentration of 1 mM of H_2_O_2_. At indicated time points, 10 μL were taken from each sample and diluted 1:200 in DPBS; 100 μL of each sample were transferred into a 96-well black plate with clear bottom (Corning). Amplex Red (Thermo Fisher) was used to detect H_2_O_2_ according to manufacturer’s instructions. Briefly, 100 μL of Amplex Red mix was added to each well for a final concentration of 27.5 μM Amplex Red and 0.1 UI/mL horseradish peroxidase (Sigma-Aldrich). The plate was incubated for 10 min at 37 °C and the fluorescence (excitation 535 nm, detection 595 nm) was read in a Spectramax Paradigm (Molecular Devices, Wokingham, UK). A H_2_O_2_ calibration curve was generated by 1:2 serial dilutions of H_2_O_2_ in DPBS (from 0.11 mM to 1.07 × 10^−4^ mM) and used to calculate the H_2_O_2_ concentration of the samples by linear regression.

### 2.8. Complementation of gpmA

The *E. coli* MG1655 *gpmA* gene with its native promoter was amplified from genomic DNA using KOD DNA polymerase (Toyobo) and the primers in Table 2 (gene ID Ecocyc database: EG11699). The single amino acid replacement of the 11^th^ histidine by an alanine (*gpmA* His11Ala) was obtained by overlap PCR using primers described in Table 2. The pWSK29 plasmid [23] was a kind gift from M. Roch (Geneva University). The plasmid and the PCR products were digested with the restriction enzymes *Eco*RI and *Kpn*I (Thermo Fisher) and were gel-purified using QIAquick gel cleanup kit (Qiagen). T4 ligase (New England Biolabs) was used for the ligations and the ligation products (pWSK29 with either *gpmA* or *gpmA* His11Ala) were transformed in TOP10 Chemically Competent *E. coli* (C404010, Thermo Fisher). The coding region of the two cloned plasmids was verified by Sanger sequencing. Plasmids were electroporated in either WT or Δ*gpmA* strains.

The complemented Δ*gpmA* strains (pWSK29 with either *gpmA* or *gpmA* His11Ala) were compared to the WT and the Δ*gpmA* strains harboring the pWSK29 (empty) plasmid on LB agar plates containing ampicillin 100 μg/mL.

### 2.9. Software

Artemis v.18.1.0 for Windows (Wellcome Sanger Institute, Cambridge, UK) was used to visualize the TraDIS data [28]. Graphpad Prism v.9.4.1 for Windows (GraphPad Software, San Diego, CA, USA) was used for data processing, graph plotting and statistical analysis. Inkscape v.1.1.1 for Windows was used for image editing (https://inkscape.org). Representation on metabolic map of previously acquired RNA-seq data was performed using the metabolism tool of Ecocyc [29,30]. All TraDIS data can be visualized at http://tradis-vault.qfab.org/.

## 3. Results

### 3.1. TraDIS Was Performed under Sublethal H_2_O_2_ Exposure

To determine the optimal dose of H_2_O_2_ to apply for the TraDIS experiment, we tested different concentrations of H_2_O_2_ on the *E.coli* strain BW25113, the strain used to generate the TraDIS library [18] (Figure 1A). The application of 2.5 mM H_2_O_2_ increased the lag phase by 70 min, whereas 5 mM or more resulted in complete absence of bacterial growth. The growth rate of bacteria during the exponential phase (between OD 0.2 and 1.6) was identical when treated with 2.5 mM compared to no treatment.

We performed the TraDIS experiment in similar conditions with 2.5 mM H_2_O_2_. The H_2_O_2_-treated condition reached OD = 1 approximatively 140 min after untreated controls. We used the genome browser Artemis to observe the insertion site of the transposons (Figure 1B). To analyze the comparative fitness of each gene under both conditions, we performed fitness analysis with the Bio::TraDIS pipeline.

In TraDIS and other Tn-seq techniques, the fitness of each gene deletion is assessed by sequencing. Mutants that are less fit in a given condition will be outcompeted and therefore less abundant, which is approximately measured by insertion frequency. To ensure the relevance of our data, we scrutinized the TraDIS data for an impact on *oxyR*. The transcription factor OxyR is a well-described H_2_O_2_ sensor that regulates *E. coli* antioxidant response and deletion of the gene has been shown to increase sensitivity against oxidative stress [31]. As expected, the frequency of insertions was significantly reduced after exposure to H_2_O_2_ indicating *oxyR* mutants are less fit than the wild type in the presence of H_2_O_2_ (Figure 1C). Using the values derived for *oxyR* as a threshold, we identified nine genes that displayed higher fold-change values suggesting of a role for each of these genes in H_2_O_2_ tolerance (Table 3). Several genes were already described in the oxidative stress response.

No transposon insertion was significantly overrepresented in the H_2_O_2_ condition, suggesting that no gene deletion is protective against H_2_O_2_ in these conditions. This analysis considered only transposon insertions inside the coding regions of genes. Transposons disrupting promoters or altering the expression of genes such as polar effect were not considered by this analysis.

### 3.2. H_2_O_2_ Susceptibility of Single-Gene Deletion Identified by TraDIS

Single-deletion mutants of genes identified by the TraDIS experiments were tested against H_2_O_2_ to evaluate the sensitivity of each mutant. To ensure the absence of undesired mutations, cognate *E. coli* strain MG1655 mutants were created by P1 phage transduction of the relevant mutations from the Keio collection [21]. The susceptibility to H_2_O_2_ of the single-gene deletion mutants was assessed by disk diffusion assay. The mutant deleted for the catalase *katG,* known as the principal H_2_O_2_ scavenger at high concentration [32], was used as positive control. As expected, the deletion of *oxyR* led to a dramatic increase of the inhibition diameter generated by H_2_O_2_ (Figure 2). The deletion of *gpmA* increased the sensitivity to H_2_O_2_ to the same extent as the *katG* deletion. Similarly, loss of *hfq* also increased significantly the sensitivity to H_2_O_2_. Other genetic deletions did not significantly alter the H_2_O_2_ susceptibility in the disk diffusion assay.

### 3.3. ΔgpmA Mutant Was More Sensitive to H_2_O_2_ but Not to Other Oxidants

Single-deletion mutants were tested against other oxidants by disk diffusion assay (Figure 3). The deletion of *oxyR* and *hfq* led to an increase of the inhibition area of a wide range of oxidants (Figure 3C,E). The deletion of *gpmA* led to a hypersensitivity to H_2_O_2_ but not to other oxidants or antibiotics (Figure 3B,E). This pattern was highly similar to the sensitivity of the Δ*katG* mutant used as positive control (Figure 2E).

The deletion mutants of the other genes identified by TraDIS were also tested against these oxidants (Appendix A Appendix A). The Δ*dksA* mutant was more sensitive to methylhydroquinone, cumene peroxide, diamide and ciprofloxacin, and the Δ*nhaA* mutant was slightly more sensitive to diamide and ciprofloxacin. This suggests that these mutants, despite no increased sensitivity to H_2_O_2_ in these conditions, were more sensitive to other oxidative stresses. Other mutants did not display significant differences compared to WT.

### 3.4. gpmA Is Upregulated by H_2_O_2_ Exposure

In a previous study, we performed a RNA-seq analysis of *E. coli* BW25113 after a 10 min exposure to a sublethal concentration (2.5 mM) of H_2_O_2_ [10]. Among the ten genes identified by TraDIS, only two genes, *dps* and *gpmA,* were significantly dysregulated by H_2_O_2_ (Figure 4A). In these settings, *gpmA* was upregulated over fourfold. As *gpmA* is part of the glycolysis reaction in *E. coli*, we extracted the transcriptomic data for the glycolysis and the TCA cycle (Appendix A Appendix A). Other enzymes from the glycolysis (*pgi*, *pfkAB*, *fbaAB*, *pgk*) were also upregulated, suggesting an increased activity of glycolysis following exposure to H_2_O_2_.

We confirmed the impact of H_2_O_2_ on *gpmA* expression in the MG1655 strain used in this study by qRT-PCR. The *gpmA* RNA was upregulated following sublethal exposure of H_2_O_2_ in a dose-dependent manner (Figure 4C). Induction of *gpmA* expression was less impressive than the catalase *katG*, a known H_2_O_2_-responsive gene (Figure 4B).

### 3.5. Catalase Activity Is Not Involved in the Increased Sensitivity of ΔgpmA to H_2_O_2_

As the Δ*gpmA* mutant displayed a similar sensitivity to oxidants compared to the Δ*katG* mutant (Figure 3E), we measured catalase expression and activity in the presence of H_2_O_2_. The Δ*gpmA* mutant did not exhibit a growth defect compared to the WT in liquid LB (Appendix A), so we first assessed the sensitivity of the *gpmA* mutant to H_2_O_2_ in liquid LB medium by counting surviving bacteria after H_2_O_2_ exposure (Figure 5A). Two hours after the addition of 2.5 mM H_2_O_2_, a 100-fold difference in the number of surviving bacteria in the *gpmA* and the *oxyR* mutant compared to the WT was observed (Figure 5B).

We measured the expression levels of the three enzymes of *E. coli* that are known to degrade H_2_O_2_, the alkyl hydroperoxide reductase encoded by *ahpC*, the catalase/hydroperoxidase HPI encoded by *katG* and the catalase HPII encoded by *katE*. We compared the WT and the Δ*gpmA* strain, in presence or in absence of H_2_O_2_ (Figure 5C). There was no significant difference in the expression of the three genes between the WT and the Δ*gpmA* strain. Upregulation of *ahpC* and *katG* was observed after the addition of 2.5 mM of H_2_O_2_ in both strains. There was no significant difference in *katE* expression after H_2_O_2_ exposure, which was expected as it is not regulated by OxyR but by RpoS and upregulated during the stationary phase of bacterial growth [33].

To test if the catalase activity was affected by the deletion of *gpmA*, we measured the degradation of 1 mM of H_2_O_2_ of the WT and the Δ*gpmA* using the H_2_O_2_-sensitive probe Amplex Red. There was no difference in H_2_O_2_ degradation between the WT and the Δ*gpmA* strains. The Δ*katG* strain, which is defective for the main H_2_O_2_ scavenger at high concentration, was unable to degrade H_2_O_2_. Altogether, this suggests that the higher sensitivity of the Δ*gpmA* to H_2_O_2_ is independent of catalase activity.

### 3.6. Other Carbon Sources Cannot Compensate the H_2_O_2_ Hypersensitivity of ΔgpmA Mutant

In LB medium, amino acids are the main source of carbon and there is virtually no glucose [34]. We wondered if the supplementation with metabolites entering the central metabolism at different levels could affect the H_2_O_2_ sensitivity of the Δ*gpmA* mutant. The addition of alternative carbon source did not significantly modify the H_2_O_2_ susceptibility of the WT or the Δ*gpmA* mutant (Figure 6A). We also tested H_2_O_2_ sensitivity in M9 minimal media with these metabolites as the only carbon source. There was no difference in the sensitivity of the WT in M9 + glucose compared to LB + glucose. The WT strain was slightly more sensitive in M9 + acetate compared to M9 + glucose. The Δ*gpmA* strain displayed a higher sensitivity in the M9 conditions compared to LB conditions. M9 plates with citrate as the sole source of carbon led to limited growth even after 48 h and were therefore not measurable. Addition of 0.5% pyruvate led to a complete disappearance of the zone of inhibition (data not shown) probably because pyruvate reacts with H_2_O_2_ to produce CO_2_, acetate and water [35].

In *Salmonella* Typhimurium, the Δ*gpmA* mutant was more susceptible to H_2_O_2_ than the WT in aerobic conditions, but not in anaerobic conditions, and the addition of the electron acceptor nitrate restored the hypersusceptibility of Δ*gpmA* [20]. We tested the H_2_O_2_ susceptibility of *E. coli* WT and Δ*gpmA* in anaerobic conditions. Interestingly, it appeared that the WT was slightly more sensitive to H_2_O_2_ in anaerobic conditions than in aerobic conditions suggesting that the exposure to oxygen protect in part against H_2_O_2_ damage. However, the Δ*gpmA* mutant did not display any difference in H_2_O_2_ sensitivity between anaerobic and aerobic conditions and the difference between the Δ*gpmA* mutant and the WT was maintained in anaerobic conditions. As *E. coli* is also able to use other electron acceptors than oxygen for respiration, we tested the addition of sodium nitrate in anaerobic conditions, but this did not change the area of inhibition induced by H_2_O_2_ compared to the anaerobic condition without nitrate (data not shown).

Altogether, we explored a potential impact of factors affecting glycolysis following H_2_O_2_ exposure, but we did not observe significant changes in conditions of low oxygen or using different carbon sources.

### 3.7. The Function of gpmA Is Necessary for H_2_O_2_ Tolerance

The Δ*gpmA* was complemented by native *gpmA* gene including its natural promoter using the low copy plasmid pWSK29. The complemented strain displayed similar H_2_O_2_ susceptibility than the WT strain (Figure 7A). A mutation previously described as to be necessary for the function of *gpmA*, namely the substitution of the histidine 11 residue by an alanine [36], resulted in restauration of the hypersensitivity to H_2_O_2_. These data suggest that the function of *gpmA* is necessary to reach the WT levels of tolerance against H_2_O_2_.

*E. coli* possess a secondphosphoglycerate mutase encoded by *gpmM*, which presents no sequence similarity with *gpmA* [37]. Contrary to *gpmA*, the expression level of *gpmM* was slightly downregulated after the addition of H_2_O_2_ in a previous RNA-seq dataset (Figure 7B). This suggests that following exposure to H_2_O_2_, *gpmA* represents the principal form of phophoglygerate mutase. We tested the Δ*gpmM* mutant for H_2_O_2_ sensitivity. Contrary to *gpmA*, the deletion of *gpmM* did not increase the sensitivity to H_2_O_2_ (Figure 7C), suggesting a possible alternative function of *gpmA* in conditions of H_2_O_2_ exposure.

## 4. Discussion

The production of H_2_O_2_ by phagocytes from the human immune system and by *Lactobacilli* species of the normal microbiota are essential for the prevention of colonization from various opportunistic pathogens. Although H_2_O_2_ effects on bacteria have been studied for years, the mechanisms by which H_2_O_2_ exerts its antimicrobial activity is still incompletely understood [14,16].

Our TraDIS analysis identified 10 mutants with fitness defect upon H_2_O_2_ exposure, implicating a role for these genes under H_2_O_2_-induced oxidative stress. Only three of the ten genes, *oxyR*, *gpmA* and *hfq*, showed a significantly higher susceptibility to H_2_O_2_ when knocked-out. This could be due to the differences in the settings between the TraDIS experiment and the disk diffusion assay. For example, the DNA-binding protein encoded by *dps* protects DNA from H_2_O_2_ damage through iron sequestration and this defense is more important in stationary phase of growth [38]. However stationary phase cultures of each knockout was treated with H_2_O_2_ in liquid medium, their respective growth was not different compared to the WT, except for the Δ*oxyR* strain (data not shown). The majority of genes we identified by TraDIS (*oxyR*, *dps*, *rpoS*, *dksA*, *hfq, polA*) have already been reported to respond to oxidative stress in *E. coli*. The transcription factor OxyR is a well described sensor of H_2_O_2_, which regulates an extensive and coordinated antioxidant transcriptional response [9,39]. The RNA polymerase subunit RpoS regulates the general stress response and was previously described to be activated by oxidative stress [40], and the deletion of this gene increases sensitivity to H_2_O_2_ [41]. The RNA polymerase accessory protein DksA senses oxidative stress through its cysteine residues and participates to the transcriptional response against oxidative stress [42]. *Hfq*, a RNA-binding protein that affects many cellular processes influences both the small RNA OxyS and the translation of *rpoS* described above *in E. coli* [43,44]. The DNA polymerase I encoded by *polA* is implicated in DNA repair and non-functional PolA increases H_2_O_2_-sensitivity [45,46]. The *polA*, *rbsR*, *dps*, *oxyR, corA, rpoS* genes were also identified in a similar experiment performed previously on *Salmonella enterica* serovar Typhimurium under sublethal H_2_O_2_ exposure [19]. The *dksA* and *nhaA* mutants, despite showing no increase in sensitivity to H_2_O_2_, were slightly more sensitive to other oxidants than WT using disk diffusion assay. Other validation experiments, such as competition assay with WT under H_2_O_2_ stress, might better reflect the TraDIS experimental conditions.

On the other hand, several genes previously identified in the literature as necessary for H_2_O_2_ tolerance were not identified by this TraDIS experiment. For example, *xthA*, whose deletion mutant is more sensitive to H_2_O_2_ [47], displayed a decreased fitness in H_2_O_2_ condition but did not reach the threshold of significance. An explanation could lie in the fact that unlike antibiotics, H_2_O_2_ is rapidly degraded by bacteria. The duration of the exposure to H_2_O_2_ performed for the TraDIS may have been insufficient to identify all genes implicated in H_2_O_2_ tolerance. Secondly, the stress was applied against pooled mutants in liquid where other mutants could provide cross-protection for susceptible mutants. For example, the catalase KatG, which is known to protect against H_2_O_2_, was not identified by TraDIS, probably because of this phenomenon. This was also the case in a previous study that used Tn-seq with H_2_O_2_ in *Salmonella* Typhimurium, where none of the catalase genes were identified [19]. Thus, our TraDIS data only identified those mutants that showed fitness defects despite cross-protection and inherent H_2_O_2_ degradation.

The TraDIS experiment also identified genes that, to our knowledge, were not previously associated with *E. coli* H_2_O_2_ sensitivity. The magnesium ion transporter encoded by *corA* had been shown to be more sensitive to lactoperoxidase–thiocyanate stress but not to H_2_O_2_ [48]. *rbsR* controls the transcription of the operon involved in ribose catabolism and transport and the salvage pathway of purine nucleotide synthesis [49]. *corA* and *rbsR* were also identified in a similar Tn-seq experiment using H_2_O_2_ on *Salmonella enterica* serovar Typhimurium [19]. The Na^+^:H+ antiporter *nhaA* is implicated in other stress responses against sodium ion, pH homeostasis and in maintaining antibiotic tolerance under starvation [50]. The glycolysis enzyme *gpmA* has been previously identified by a Tn-seq experiment following H_2_O_2_ exposure in the Gram-negative bacteria *Salmonella enterica* serovar Typhimurium [20].

When tested with diverse oxidants that damage bacteria through different modes of action, Δ*gpmA* was specifically more sensitive to H_2_O_2_, like the Δ*katG* strain. However, it was not through a differential expression of H_2_O_2_-scavenging genes or a decreased catalase activity of the strain, suggesting a different mode of action. Moreover, the upregulation of *gpmA* by sublethal exposure of H_2_O_2_ suggests the importance of *gpmA* in H_2_O_2_ tolerance. Under oxidative stress, some enzymes of the central metabolism have been shown to be upregulated. The glucose-6-phosphate isomerase encoded by *pgi* have been shown to be regulated by the oxidative stress sensitive regulators SoxRS [51]. Similarly, in the TCA cycle, the aconitase *acnA* and the fumarase *fumC* are regulated by SoxRS and are upregulated under H_2_O_2_ exposure [52,53]. The hypersensitivity to H_2_O_2_ of Δ*gpmA* mutant could be complemented with the low-copy plasmid pWSK29 expressing the WT *gpmA* gene under its native promoter but not if the histidine 11 was mutated to an alanine. This strongly suggests that the function of *gpmA* affects *E. coli* tolerance to H_2_O_2_.

Surprisingly, addition of other metabolites or the absence of oxygen did not abolish the difference in H_2_O_2_ sensitivity between the WT and the Δ*gpmA* mutant. These data contrast with previous work on *Salmonella enterica* serovar Typhimurium, where other metabolites entering metabolism downstream of *gpmA* reaction (for a scheme of glycolysis, see Appendix A) could complement the increased sensitivity of a Δ*gpmA* mutant and where anoxic environment abolished the difference of H_2_O_2_ susceptibility between WT and Δ*gpmA* mutant [20]. In the same study, metabolomics approach showed that H_2_O_2_ exposure led to an increase of glycolysis and fermentation that was important in *Salmonella* H_2_O_2_ tolerance. This contrasts with previous metabolomics analysis on *E. coli* after H_2_O_2_ treatment which reported a decrease of metabolites related to glycolysis and TCA cycle, changes that were common to other stress conditions such as heat shock and cold stress [54]. Altogether, this suggests a different metabolic adaptation to H_2_O_2_ stress between *E. coli* and *Salmonella* and a difference of *gpmA* function. More research is needed to better understand the mechanisms of *gpmA* effects in H_2_O_2_ tolerance in *E. coli* and in other organisms.

Contrary to vertebrates that only possess one phosphoglycerate mutase, some eubacteria, among which relevant pathogens including *E. coli*, encode two enzymes that display no sequence similarity [55]. The 2,3-bisphosphoglycerate-dependent phosphoglycerate mutase (or dPGM), encoded by *gpmA* is common to bacteria and vertebrates, whereas 2,3-bisphosphoglycerate-independent phosphoglycerate mutase (or iPGM) encoded by *gpmM* is shared by bacteria and higher plants. As the double deletion of *gpmA* and *gpmM* have been suspected non-viable in *E. coli*, the glycolysis function is assumed by *gpmM* in the *gpmA*-deleted strain and vice versa [56]. The deletion of *E. coli gpmM* did not affect H_2_O_2_ sensitivity, suggesting that only *gpmA* function has a role under H_2_O_2_ exposure. This led to the hypothesis that *gpmM* could be damaged by H_2_O_2_ and its function is replaced by *gpmA* under H_2_O_2_ exposure. This happens for other enzymes of the TCA cycle, the aconitase and the fumarase, where oxidative-resistant isoforms (*acnA*, *fumC*) replace oxidative-sensitive isoforms (*acnB*, *fumA*, *fumB*), after H_2_O_2_ exposure [52,53]. Cysteine residues can be more prone to oxidation by H_2_O_2_ than other amino acids [57]. GpmM possesses two cysteine residues, which can result in H_2_O_2_-induced damage from oxidation of these residues. As GpmA does not possess cysteine residues, it could be more resistant to H_2_O_2_ than GpmM. The cysteine residues of GpmM are not implicated in active sites described in current models (Ecocyc, Uniprot). While Cys397 seems buried and is not conserved in Gram-positive bacteria, Cys424 seems to be more accessible on the protein models and is present in both Gram-negative *(P. aeruginosa, Salmonella enterica, K. pneumoniae*) and Gram-positive bacteria (*S. aureus*, *B. subtilis*). Additional studies are needed to evaluate their potential implication in oxidative stress susceptibility.

## 5. Conclusions

This work was aimed at expanding the knowledge of which genes are implicated in H_2_O_2_ tolerance. The main finding of this study was that a functional *gpmA* gene is required for tolerance to H_2_O_2_. This is the first time that *gpmA* was highlighted as an important contributor to the *E. coli* tolerance to H_2_O_2_, and it links defense against oxidative stress to central metabolism.

## Figures and Tables

**Figure 1 antioxidants-11-02053-f001:**
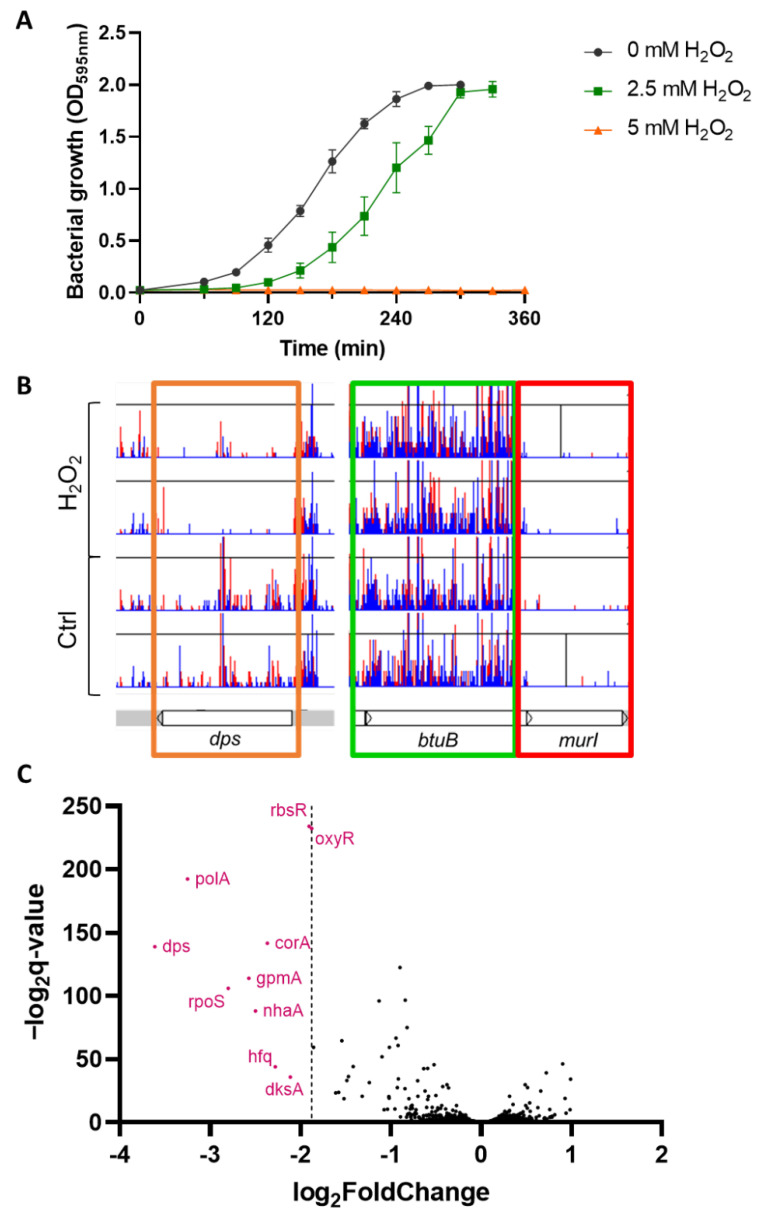
Transposon-directed insertion sequencing (TraDIS) screen of *E.coli* BW25113 under sublethal concentration of H_2_O_2_ (**A**) Bacterial growth of *E. coli* BW25113 over time with 2.5, 5 mM H_2_O_2_ and untreated control. The concentration of 2.5 mM was chosen for TraDIS experiment as it displayed a high reproducibility and a delay of approximatively 70 min compared to control (mean +/− SD, *N* = 3); (**B**) Visualization of the TraDIS data using Artemis where each sample is represented by a histogram depicting the localization (X-axis) and the frequency (Y-axis) of transposon insertion sites (in red are antisense insertions, in blue are same sense insertions). Representative examples of essential gene (*murI* in red box), non-essential genes (*btuB*, in green box) and genes with a reduced fitness in the H_2_O_2_ condition (*dps* in orange box), (*N* = 2) (**C**) Fitness analysis of the TraDIS data, H_2_O_2_-treated condition compared to control. Each dot represents a gene, X-axis represents the difference in number of insertions in the H_2_O_2_ condition compared to control, Y-axis represents the statistical significance. Nine genes (in pink) displayed a significant and more extreme change than the H_2_O_2_-sensor gene *oxyR*.

**Figure 2 antioxidants-11-02053-f002:**
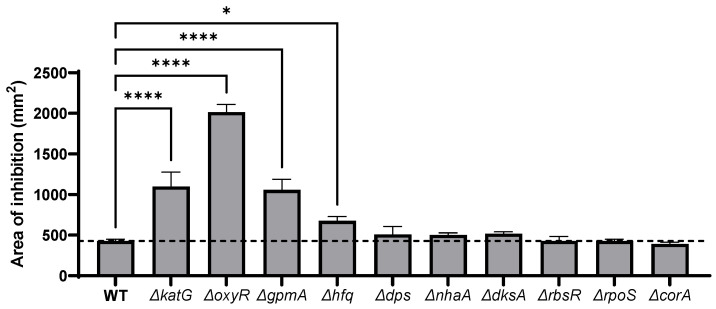
Sensitivity to H_2_O_2_ of genes identified by TraDIS. The sensitivity to 10 μL of 1 M H_2_O_2_ of each single-gene deletion mutant identified by the TraDIS was assessed by disk diffusion assay. The coding regions of *E. coli* MG1655 (WT) were replaced by the kanamycin cassette of the corresponding mutant originated from the Keio collection using the phage P1 to ensure the absence of undesired mutation. Data were analyzed by one-way ANOVA with Tukey test for multiple comparison and *, **** correspond to *p* < 0.05 and 0.0001 respectively (mean +/− SD, *N* = 3).

**Figure 3 antioxidants-11-02053-f003:**
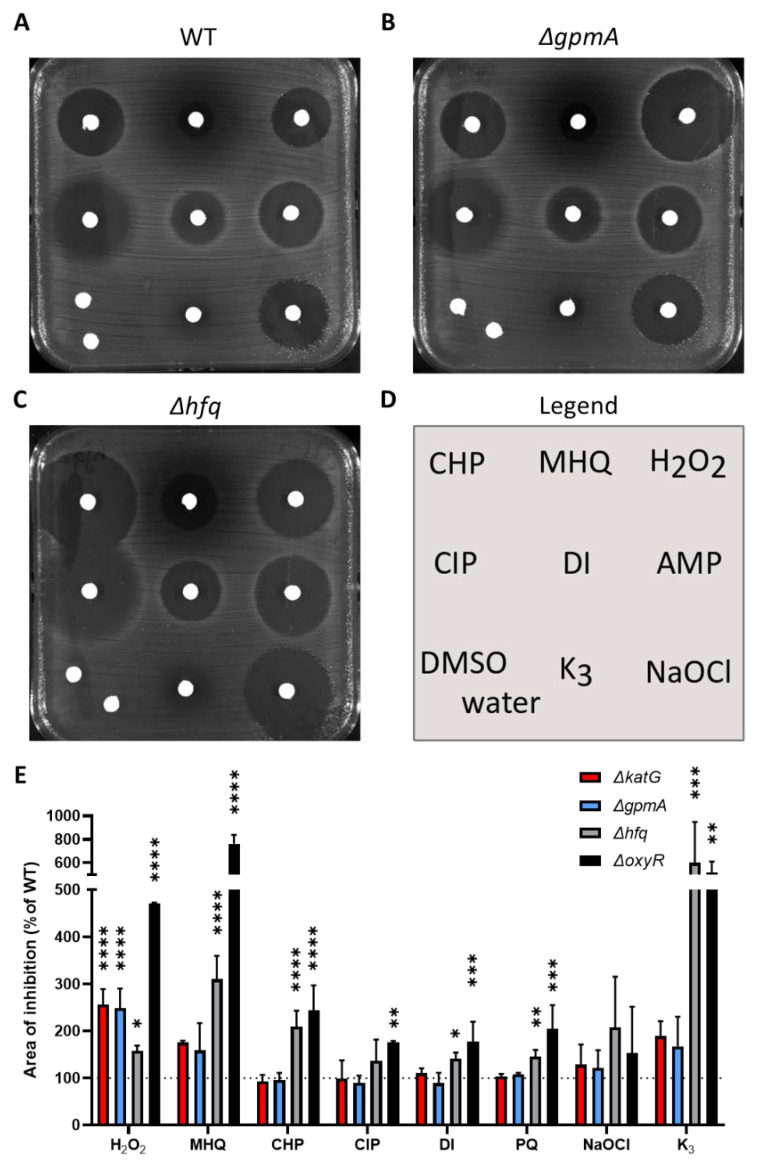
Sensitivity of the Δ*oxyR*, Δ*gpmA* and Δ*hfq* mutants exposed to various oxidants. (**A**) WT, (**B**) Δ*gpmA*, (**C**) Δ*hfq.* (**D**) Oxidants applied on each disk (CHP: cumene hydroperoxide, MHQ: methylhydroquinone, H_2_O_2_: hydrogen peroxide, CIP: ciprofloxacin, DI: diamide, AMP: ampicillin, K3: menadione, NaOCl: sodium hypochlorite, DMSO: dimethylsulfoxide). (**E**) Quantification of the area of inhibition normalized to WT for each oxidant. One-way ANOVA with Tukey multiple comparison was performed separately for each oxidant on the area of inhibition of the WT, Δ*katG* and the 9 mutants identified by TraDIS (Appendix A). The significance of the difference with the WT is represented on the normalized data by stars (mean +/− SD, *N* = 3). *, **, ***, **** correspond to *p* < 0.05, 0.01, 0.001, and 0.0001, respectively.

**Figure 4 antioxidants-11-02053-f004:**
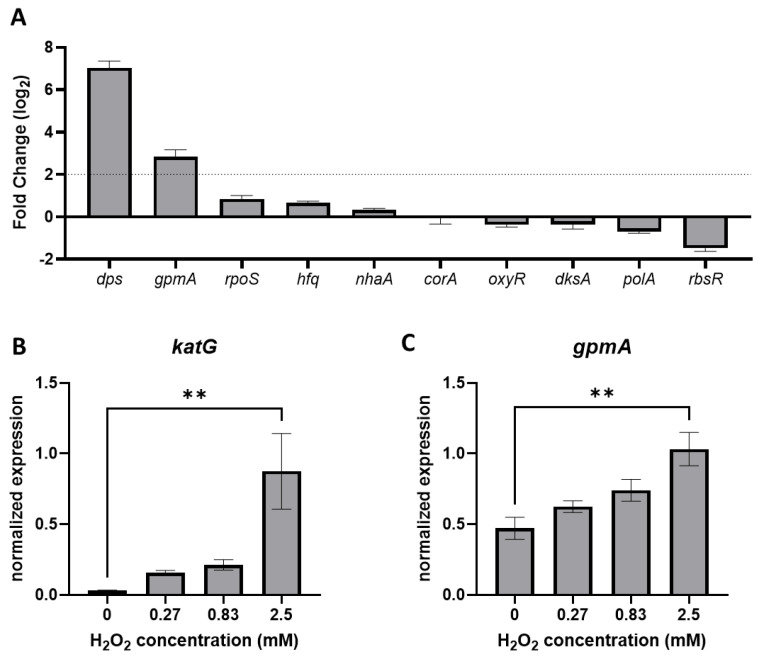
*gpmA* expression is upregulated following exposure to H_2_O_2_. (**A**) Differential expression of the 10 genes identified by TraDIS 10 min after exposition to 2.5 mM H_2_O_2_ compared to no treatment (mean +/− SD, *N* = 4). Data originated from previously performed RNA-seq (deposited on ENA with the accession number: PRJEB51098) [10]. (**B**,**C**) Levels of expression of *katG* and *gpmA,* respectively, in the strain MG1655 under increasing concentration of H_2_O_2_, assessed by qRT-PCR, ** corresponds to *p* < 0.01. (SEM +/− SD, *N* = 3).

**Figure 5 antioxidants-11-02053-f005:**
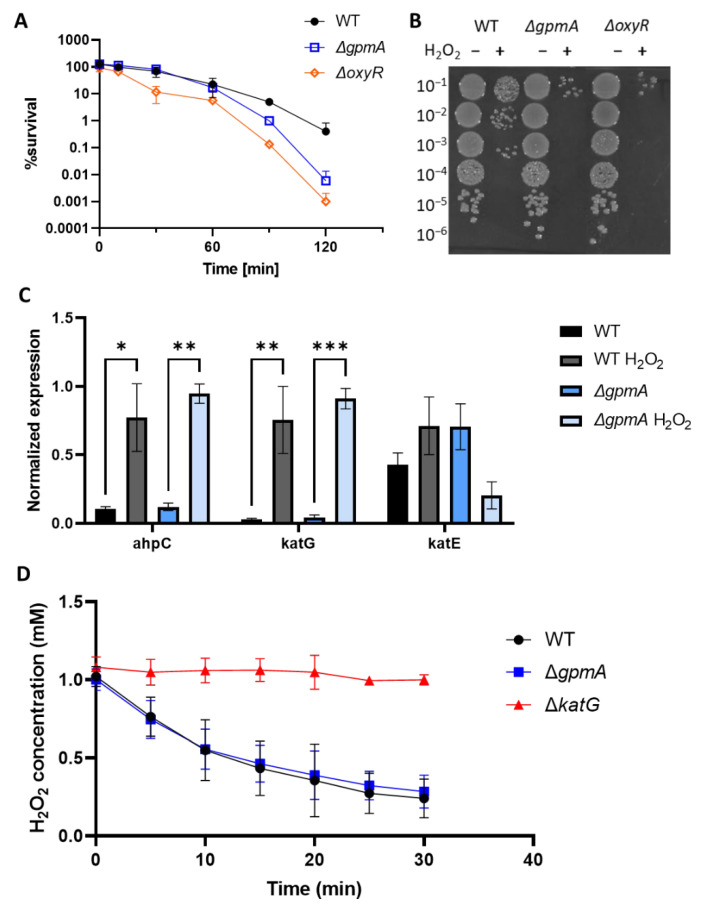
WT and *gpmA* displayed no difference in catalase expression and activity. (**A**) Survival of 2 × 10^7^ cells of WT (black), Δ*gpmA* (blue) and Δ*oxyR* (orange) over time after an exposure to 2.5 mM H_2_O_2_ in liquid LB (mean +/− SD, *N* = 3). *(***B**) Representative image of the survival 2h after H_2_O_2_ treatment. Each spot represents 10 μL at the given dilution factor. (**C**) Expression levels of *ahpC*, *katG* and *katE* in WT (black) or Δ*gpmA* (blue) 10 min after the addition of 2.5 mM H_2_O_2_ or corresponding control (mean +/− SEM, *N* = 3). Data were analyzed by Welch T-test, and *, **, *** correspond to *p* < 0.05, 0.01, 0.001, respectively. (**D**) Degradation of 1 mM H_2_O_2_ over time by WT (black), Δ*gpmA* (blue) and Δ*katG* (red) assessed by Amplex Red (mean +/− SD, *N* = 3).

**Figure 6 antioxidants-11-02053-f006:**
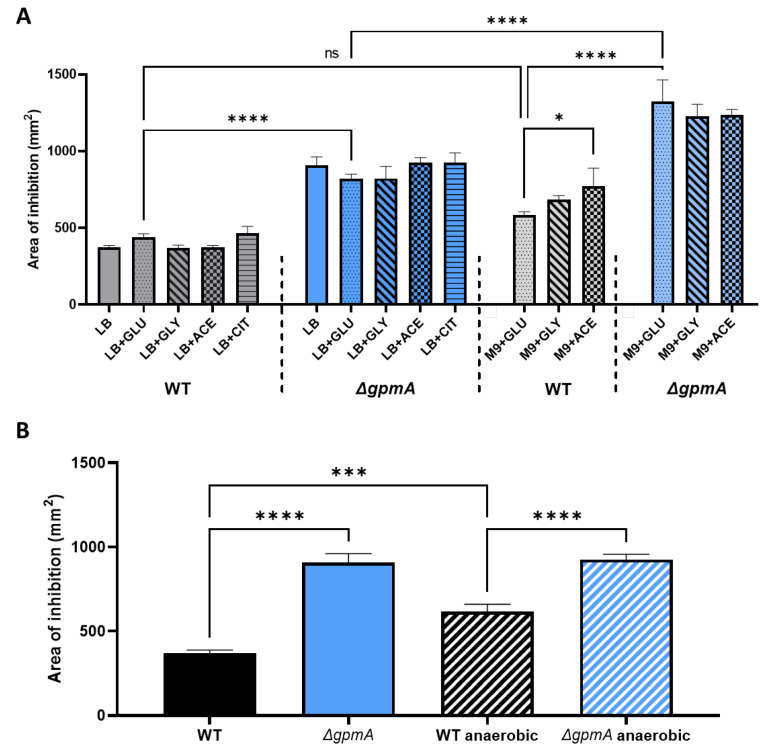
The difference in H_2_O_2_ tolerance between WT and Δ*gpmA* mutant is not affected by the addition of other carbon sources or the absence of oxygen. (**A**) Area of inhibition assessed by disk diffusion assay of WT and Δ*gpmA* strain on LB and M9 minimal medium complemented with diverse carbon source. GLU: glucose, GLY: glycerol, ACE: acetate, CIT: citrate (mean +/− SD, *N* = 3). (**B**) Area of inhibition of the WT and Δ*gpmA* strain under aerobic and anaerobic conditions (mean +/− SD, *N* = 3). Data in (**A**,**B**) were analyzed by one-way ANOVA with Tukey test for multiple comparison, and *, ***, **** correspond to *p* < 0.05, 0.001, 0.0001 respectively.

**Figure 7 antioxidants-11-02053-f007:**
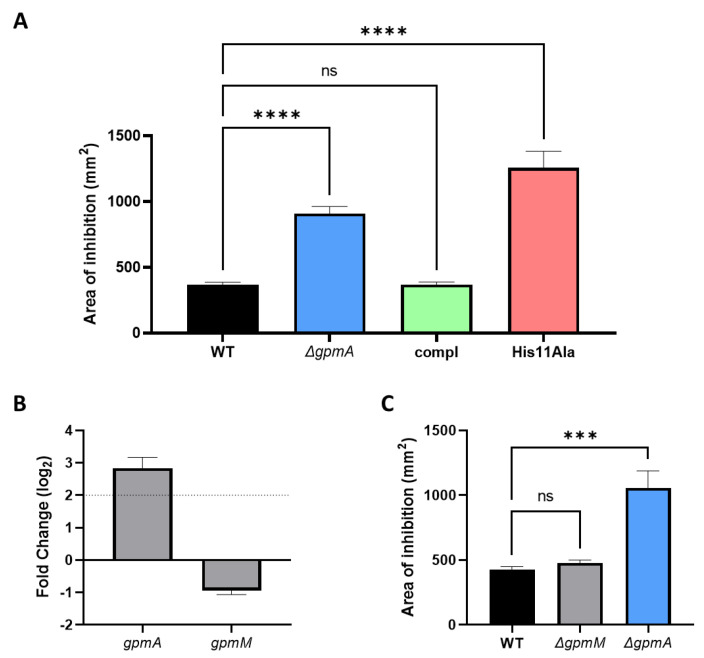
The function of *gpmA,* but not *gpmM*, is necessary to reach WT level of H_2_O_2_ tolerance. (**A**) Sensitivity to H_2_O_2_ assessed by disk diffusion assay for the WT with the empty plasmid, the Δ*gpmA* mutant with the empty plasmid, the Δ*gpmA* mutant with the plasmid encoding the native sequence of *gpmA* (compl) and the Δ*gpmA* mutant complemented with the plasmid encoding *gpmA* with the replacement of the histidine 11 by an alanine (His11Ala). All tests were performed in LB + ampicillin. Data were analyzed by one-way ANOVA with Tukey test for multiple comparison and **** correspond to *p* < 0.0001 (mean +/− SD, *N* = 3). (**B**) Differential expression of *gpmA* and *gpmM* 10 min after exposition to 2.5 mM H_2_O_2_ compared to no treatment (mean +/− SD, *N* = 4). Data from previously performed RNA-seq (deposited on ENA with the accession number: PRJEB51098) [10]. (**C**) Sensitivity to H_2_O_2_ assessed by disk diffusion assay of WT, Δ*gpmA* and Δ*gpmM* mutants. Data were analyzed by Welch *t*-test and *** correspond to *p* < 0.001 (mean +/− SD, *N* = 3).

**Table 1 antioxidants-11-02053-t001:** Bacterial strains and plasmid used in this study.

Name	Genotype	Source or Reference
BW25113	F-, ∆(*araD*-*araB*)*567*, ∆l*acZ*4787(::rrnB-3), *rph-1,* ∆(*rhaD-rhaB*)*568, hsdR514*	CGSC ^1^ [21]
MG1655	F*-*, *λ^−^*, *rph-1*	CGSC ^1^
BEFB02	MG1655, Δ*oxyR*::Cm^r^	[22]
JW3914	BW25113, Δ*katG::kan*	[21]
Δ*katG*	MG1655, Δ*katG::kan*	This study
JW0738	BW25113, Δ*gpmA::kan*	[21]
Δ*gpmA*	MG1655, Δ*gpmA::kan*	This study
JW4130	BW25113, Δ*hfq::kan*	[21]
Δ*hfq*	MG1655, Δ*hfq::kan*	This study
JW0797	BW25113, Δ*dps::kan*	[21]
Δ*hfq*	MG1655, Δ*dps::kan*	This study
JW3789	BW25113, Δ*corA::kan*	[21]
Δ*hfq*	MG1655, Δ*corA::kan*	This study
JW5437	BW25113, Δ*rpoS::kan*	[21]
Δ*rpoS*	MG1655, Δ*rpoS::kan*	This study
JW3732	BW25113, Δ*rbsR::kan*	[21]
Δ*rbsR*	MG1655, Δ*rbsR::kan*	This study
JW0141	BW25113, Δ*dksA::kan*	[21]
Δ*dksA*	MG1655, Δ*dksA::kan*	This study
JW0018	BW25113, Δ*nhaA::kan*	[21]
Δ*nhaA*	MG1655, Δ*nhaA::kan*	This study
JW3587	BW25113, Δ*gpmM::kan*	[21]
Δ*gpmM*	MG1655, Δ*gpmM::kan*	This study
pWSK29	AmpR	[23]

^1^ *E. coli* Genetic Stock Center.

**Table 2 antioxidants-11-02053-t002:** Primers used in this study.

Name	Sequence	Gene Accession ID Ecocyc	Efficiency (RT-qPCR Primers)	Reference
**RT-qPCR primers**				
gyrB_N_qPCR_F	GTCCTGAAAGGGCTGGATG	EG10424	1.89 (89.37%)	[27]
gyrB_N_qPCR_R	CGAATACCATGTGGTG-CAGA			
gyrB_V_qPCR_F	GAAATTCTCCTCCCAGACCA	EG10424	1.83 (82.56%)	[27]
gyrB_V_qPCR_R	GCAGTTCGTTCATCTGCTGT			
katG_qPCR_F	GGGCCGACCTGTTTATCCTC	EG10511	1.92 (92.09%)	[10]
katG_qPCR_R	ATCCAGATCCGGTTCCCAGA			
gpmA_qPCR_F	AGCCATGCCTGATCCAGTTC	EG11699	2.00 (100.45%)	This study
gpmA_qPCR_R	TTTCACCGGTTGGTACGACG			
hfq_qPCR_F	CTACTGTTGTCCCGTCTCGC	EG10438	2.01 (101.14%)	This study
hfq_qPCR_R	TCGGTTTCTTCGCTGTCCTG			
ahpC_qPCR_F	TGCGACCTTCGTTGTTGACC	EG11384	2.00 (100.23%)	This study
ahpC_qPCR_R	CGGAGCCAGAGTTGCTTCAC			
katE_qPCR_F	TCCGGAATACGAACTGGGCT	EG10509	2.08 (108.44%)	This study
katE_qPCR_R	ATTTTGCCGACACGCTGAAC			
**Cloning primers for *gpmA* (EG11699)**				
pWSK_gpmA_KpnI_R	GGGGTACCCCGACGTTTACTTCGCTTTACCCTG			This study
pWSK_EcoRI_gpmA_F	GGAATTCCATCACCAGCAAACACCGAC			This study
gpmA_His11Ala_F	CTGGTTCTGGTTCGTGCGGGCGAAAGTCAG			This study
gpmA_His11Ala_R	CTGACTTTCGCCCGCACGAACCAGAACCAG			This study

**Table 3 antioxidants-11-02053-t003:** Genes underrepresented in the H_2_O_2_ condition of the TraDIS experiment. The gene function and the fold change compared to control are detailed.

Gene Name	Function	Log_2_ FC	q Value
** *corA* **	magnesium/nickel/cobalt transporter	−2.37	2.11 × 10^−43^
** *dksA* **	transcriptional regulator of rRNA transcription, DnaK suppressor protein	−2.11	1.56 × 10^−11^
** *dps* **	Fe-binding and storage protein; stress-inducible DNA-binding protein	−3.61	1.37 × 10^−42^
** *gpmA* **	phosphoglyceromutase 1	−2.57	4.52 × 10^−35^
** *hfq* **	global sRNA chaperone; HF-I, host factor for RNA phage Q beta replication	−2.28	5.73 × 10^−14^
** *nhaA* **	sodium-proton antiporter	−2.50	2.71 × 10^−27^
** *oxyR* **	oxidative and nitrosative stress transcriptional regulator	−1.88	1.17 × 10^−70^
** *polA* **	fused DNA polymerase I 5′->3′ polymerase/3′->5′ exonuclease/5’->3’ exonuclease	−3.25	1.29 × 10^−58^
** *rbsR* **	transcriptional repressor of ribose metabolism	−1.90	3.27 × 10^−71^
** *rpoS* **	RNA polymerase, sigma S (sigma 38) factor	−2.80	1.11 × 10^−32^

## Data Availability

The data for this study have been deposited in the European Nucleotide Archive (ENA) at EMBL-EBI under accession number PRJEB56340 (https://www.ebi.ac.uk/ena/browser/view/PRJEB56340). Processed data are available for viewing at our online browser: https://tradis-vault.qfab.org/.

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
