# Peer review of "Transposon-Directed Insertion-Site Sequencing Reveals Glycolysis Gene gpmA as Part of the H2O2 Defense Mechanisms in Escherichia coli"

_antioxidants, 2022, doi:10.3390/antiox11102053_

Round 1
Reviewer 1 Report
The presented study is aimed at the actual problem of pathogenic bacteria defence against peroxide. The research was performed with strains of E.coli, and the authors used a rich arsenal of modern genetic and molecular biological methods. All experimental procedures were fully described. The study is of interest to microbiology professionals. The language is sufficient to understand the content. However, some corrections need to be made to the text as listed below. In addition, I have several questions for the authors (see below). Note the following designations: <...> - for inclusion and - for deletion:
547: The next phrase in the Conclusions is puzzling: "The main finding of this study was that a functional gpmA enzyme is required for tolerance to H2O2." First, there are no experiments with functional enzymes in the study presented. Second, the designation "gpmA enzyme" is used noncorrectly in this phrase. Such designations are used for genes in the scientific literature. Third, the results obtained by the authors clearly demonstrate that in the absence of katG (mutant DkatG) but in the presence of gpmA, E.coli performs no function of H2O2 degradation (Figure 5, D). This fact is not discussed in the text for some unknown reason. It would be very interesting to discuss this result and to present available explanations. It seems obvious that the product of gpmA has no direct peroxidase function. What do the authors think?
Suggested corrections to the text:
Introduction
29: It <is> a frequent
51: <through> the oxidation
52: allowing <to regulate>
66: results of <our> study
Methods
129: <susceptibility>
148, 187: <1> mL
152: <10> μL
160: The phrase is not clear. Did you mean the following? : The sample was <diluted> in 2 mL <of LB>
174: <2 min>
175: <10 min>, <for> 15 s, <for 1 min>
188: < bacterial suspension>
194: <for> 10 min
Results
276: <as> the katG
293: Figure 3.<B>
312: <min>
327, 366: <10 min>
page 11, Figures 4, B and C, inscriptions under absciss: <[H2O2], (mM)>
page 11, Figures 4, B and C, inscriptions on the top: katG, gpmA < katG>, <gpmA>
321: <(Figure 4.C)
322: (Figure 4.C) <(Figure 4.B)
340: in liquid <LB>
ÖDiscussion
462: <previously>
474: was applied against pooled mutants <in liquid>
475: could <provide>
480: < H2O2 degradation>
505-506: gpmA <affects>
519-520: < More research is needed> to better understand the mechanisms of gpmA <effects>
536: GpmM <possesses> 2 cysteine residues, <which can result in> H2O2-induced <damage from> oxidation of these residues.
540: < current>
540: <While> Cys397 seems buried and is not conserved in Gram-positive bacteria, Cys424 <seems> to be more accessible in protein models
546: This work <was> aimed<at> expanding the knowledge ...
Author Response
Comments and Suggestions for Authors
The presented study is aimed at the actual problem of pathogenic bacteria defence against peroxide. The research was performed with strains of E.coli, and the authors used a rich arsenal of modern genetic and molecular biological methods. All experimental procedures were fully described. The study is of interest to microbiology professionals. The language is sufficient to understand the content. However, some corrections need to be made to the text as listed below. In addition, I have several questions for the authors (see below). Note the following designations: <...> - for inclusion and - for deletion:
AU: We thank the reviewer for the general positive evaluation of our manuscript and its constructive comments. Please find below a point-to-point response to the questions raised by reviewer #1
547: The next phrase in the Conclusions is puzzling: "The main finding of this study was that a functional gpmA enzyme is required for tolerance to H2O2."
First, there are no experiments with functional enzymes in the study presented. Second, the designation "gpmA enzyme" is used noncorrectly in this phrase. Such designations are used for genes in the scientific literature.
AU: We agree with reviewer #1. The sentence was modified to “The main finding of this study was that a functional gpmA gene is required for tolerance to H2O2”
Third, the results obtained by the authors clearly demonstrate that in the absence of katG (mutant DkatG) but in the presence of gpmA, E.coli performs no function of H2O2 degradation (Figure 5, D). This fact is not discussed in the text for some unknown reason. It would be very interesting to discuss this result and to present available explanations. It seems obvious that the product of gpmA has no direct peroxidase function. What do the authors think?
AU: The mechanism of action of GpmA under H2O2 exposure is unknown. Since the sensitivity to H2O2 was similar between ΔgpmA and ΔkatG mutants and similarly specific to H2O2 over other oxidants, we hypothesized that ΔgpmA mutant might have a deficit in catalase activity or levels. The RNA expression of known H2O2 scavengers was comparable in WT and ΔgpmA mutant both at baseline and after induction by a sublethal concentration of H2O2. Measurement of H2O2 degradation showed that a fully active catalase activity was present in the ΔgpmA mutant. This is stated in the results between lines 360-365. The discussion contains the following sentence: “When tested with various oxidants that damage bacteria through different modes of action, ΔgpmA was specifically more sensitive to H2O2, like the ΔkatG strain. However, it was not through a differential expression of H2O2-scavenging genes or a decreased catalase activity of the strain, suggesting a different mode of action”.
Suggested corrections to the text:
Introduction
29: It <is> a frequent
51: <through> the oxidation
52: allowing <to regulate>
66: results of <our> study
Methods
129: <susceptibility>
148, 187: <1> mL
152: <10> μL
160: The phrase is not clear. Did you mean the following? : The sample was <diluted> in 2 mL <of LB>
174: <2 min>
175: <10 min>, <for> 15 s, <for 1 min>
188: < bacterial suspension>
194: <for> 10 min
Results
276: <as> the katG
293: Figure 3.<B>
312: <min>
327, 366: <10 min>
page 11, Figures 4, B and C, inscriptions under absciss: <[H2O2], (mM)>
page 11, Figures 4, B and C, inscriptions on the top: katG, gpmA < katG>, <gpmA>
321: <(Figure 4.C)
322: (Figure 4.C) <(Figure 4.B)
340: in liquid <LB>
Discussion
462: <previously>
474: was applied against pooled mutants <in liquid>
475: could <provide>
480: < H2O2 degradation>
505-506: gpmA <affects>
519-520: < More research is needed> to better understand the mechanisms of gpmA <effects>
536: GpmM <possesses> 2 cysteine residues, <which can result in> H2O2-induced <damage from> oxidation of these residues.
540: < current>
540: <While> Cys397 seems buried and is not conserved in Gram-positive bacteria, Cys424 <seems> to be more accessible in protein models
546: This work <was> aimed<at> expanding the knowledge ...
AU: We thank reviewer #1 for his careful reading. We modified the manuscript accordingly.
Reviewer 2 Report
This manuscript describes identification of gpmA as the gene responsible for H2O2 defense in E. coli. They have performed a Transposon-mediated genetic analyses of genes required for H2O2 defense, and identified several genes involved in the defense. They have confirmed the involvement of gmpA through several experiments.
On the whole, the research is well performed and the ms is well written.
Only several minor points
P9l29 figure C and E should be corrected to 3C and 3E
The reason why they focused on gpmA out of nine genes should be described more in detail.
Other genes should be described in the abstract.
Author Response
Comments and Suggestions for Authors
This manuscript describes identification of gpmA as the gene responsible for H2O2 defense in E. coli. They have performed a Transposon-mediated genetic analyses of genes required for H2O2 defense, and identified several genes involved in the defense. They have confirmed the involvement of gmpA through several experiments.
On the whole, the research is well performed and the ms is well written.
AU: We would like to thank reviewer #2 for the positive comments about our work.
Only several minor points
P9l29 figure C and E should be corrected to 3C and 3E
AU: This point was corrected.
The reason why they focused on gpmA out of nine genes should be described more in detail.
AU: We focused on gpmA because the KO has the most pronounced susceptibility to H2O2 (beside the known regulator oxyR) and this hypersuceptibility was specific to H2O2. Also, the implication of gpmA in H2O2 was not reported in E.coli yet.
Other genes should be described in the abstract.
AU: We added the TraDIS results of other genes in the abstract.